# Population processes in cyber system variability

**Marc Mangel** [1,2,3]⊙*, **Alan Brown**[4]⊙

**1** Department of Applied Mathematics, University of California, Santa Cruz, Santa Cruz, CA, United States of America, **2** Department of Biology, University of Bergen, Bergen, Norway, **3** Puget Sound Institute, University of Washington Tacoma, Tacoma, WA, United States of America, **4** Johns Hopkins University Applied Physics Laboratory, Laurel, Maryland, United States of America

⊙ These authors contributed equally to this work.
* msmangel@ucsc.edu

**Data Availability Statement:** All relevant data are within the paper and its Supporting information files.

**Funding:** MM was supported by a consulting contract with the Johns Hopkins University Applied Physics Laboratory (151015) The funders had no

## Abstract

Variability is inherent to cyber systems. Here, we introduce ideas from stochastic population biology to describe the properties of two broad kinds of cyber systems. First, we assume that each of $N_0$ components can be in only one of two states: functional or nonfunctional. We model this situation as a Markov process that describes the transitions between functional and nonfunctional states. We derive an equation for the probability that an individual cyber component is functional and use stochastic simulation to develop intuition about the dynamics of individual cyber components. We introduce a metric of performance of the system of $N_0$ components that depends on the numbers of functional and nonfunctional components. We numerically solve the forward Kolmogorov (or Fokker–Planck) equation for the number of functional components at time $t$, given the initial number of functional components. We derive a Gaussian approximation for the solution of the forward equation so that the properties of the system with many components can be determined from the transition probabilities of an individual component, allowing scaling to very large systems. Second, we consider the situation in which the operating system (OS) of cyber components is updated in time. We motivate the question of OS in use as a function of the most recent OS release with data from a network of desktop computers. We begin the analysis by specifying a temporal schedule of OS updates and the probability of transitioning from the current OS to a more recent one. We use a stochastic simulation to capture the pattern of the motivating data, and derive the forward equation for the OS of an individual computer at any time. We then include compromise of OSs to compute that a cyber component has an unexploited OS at any time. We conclude that an interdisciplinary approach to the variability of cyber systems can shed new light on the properties of those systems and offers new and exciting ways to understand them.

## Introduction

Although we may wish otherwise, variability is a constitutive property of cyber systems [1–5]. Whatever the source, cyber components that have been corrupted and are being restored to

role in study design, data collection and analysis, decision to publish, or preparation of the manuscript.

**Competing interests:** The authors have declared that no competing interests exist.

service are temporarily unable to contribute to the function of the full cyber system or the physical system that it enables [6].

Because of corruption and return to service, we should not expect individual components of a cyber system to be at a steady state, but to fluctuate between being functional and non-functional (e.g., from natural variability), compromised or not (e.g., from malware attacks), or intentionally changed (as in, the regular updating of the operating systems [OSs] of the cyber components). However, because cyber systems are collections of individual components, the collection may have steady state properties even when individual components are not in a steady state but are fluctuating. The stable performance of the system will then depend on the steady state distribution of the components rather than the status of individual components.

The questions that arise in the operational setting for cyber systems also occur in the biology of populations [7, 8], the physics of many body systems [9, 10], and communication and network-on-chip engineering [2, 11–13]. Here, we use ideas from stochastic population theory [7, 14–16] and statistical physics [17, 18] to characterize the variability of cyber systems and construct compact mathematical models for such complex dynamics [19].

Perhaps the most important notion from these fields is that we should not expect deterministic steady states of individual components (individuals in populations/atoms or molecules in physical systems), but that we can characterize temporal and steady state distributions of the system based on transition probabilities.

## Overview of the paper

We now provide a brief overview of the paper, including visual representations in Figs 1 and 2 of the problems that we address. We conclude this section with a discussion of our choice of discrete time models and how one can connect continuous and discrete time models.

## Systems with only two possible states

For components that can be in only one of two states, we model the behavior of an individual component as a two-state Markov process so that an individual component is either functional (F) or nonfunctional (NF), with the initial number of components in each state specified (see Discussion for extensions from two to multiple states). Even if all components are initially F, at any subsequent time the entire system will have a distribution for the number of F and NF components.

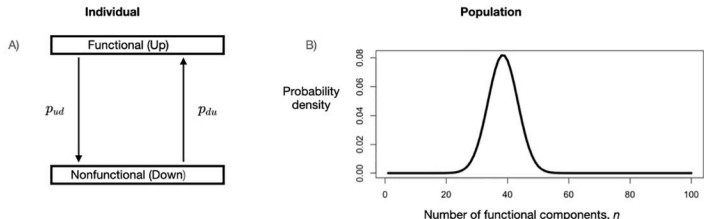

**Fig 1. Visual description of the problem when the system has only two possible states.** An individual cyber component can be either functional ("up") or nonfunctional ("down"). A) Individual components transition between the two states according to a Markov process in which $p_{ud}$ is the probability that an individual component that is currently functional is nonfunctional at the next time step and $p_{du}$ is the probability that a component that is currently nonfunctional is functional at the next time step. B) Although individual components will in general continue to fluctuate between functional and nonfunctional states, after a sufficient length of time a population of components will settle into a probability distribution for the number of functional components. Our goal is to characterize this probability distribution and the factors determining how rapidly it is approached.

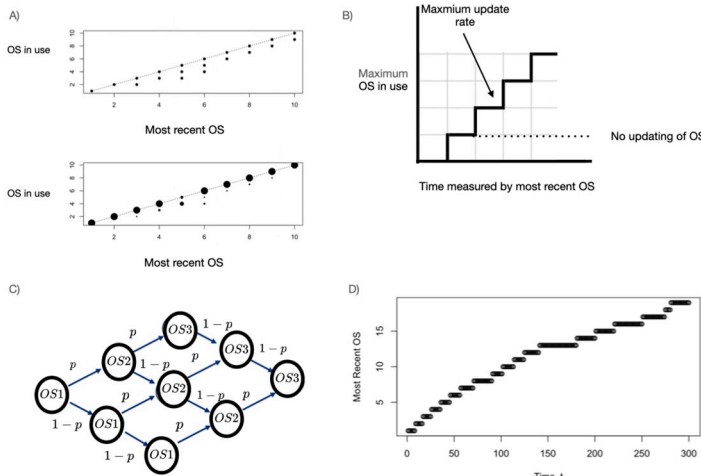

**Fig 2. Visual description of the problem when the cyber components have multiple possible OSs that are updated as time goes forward and cyber components do not return to a previous OS.** A) Two representations of the OS in use (y-axis) in a subset of about 335 computers from a network of about 7000 computers between July 2019 and March 2020 (details on how the data were chosen can be found in S1 Appendix in S1 File) as a function of the most recent OS release (x-axis). In the upper panel, we show the data as presence/absence of an OS in the sample. In the lower panel, we show the relative distribution of the current OS, where the diameter of the circle is proportional to the base-10 logarithm of the counts of that OS. The points (5,3) and (10,9) from the upper panel are not visible in the lower panel because their relative representation is so small. B) When measured by the most recent OS, time moves forward in uniform steps (x-axis), and the maximum OS in use jumps whenever a new OS is released. The maximum updating rate is then the black solid line; if an OS is never updated, it remains at the first release (dotted line). Each location between these two lines corresponds to a probability of the OS in use given the most recent OS. C) With time measured in OS release, the space between the two lines in panel B can be filled using a binomial lattice model (*sensu* Leisen and Reimer [22]) in which the current OS, at the time of an OS update, is either updated by 1 or remains the same. We let $p$ denote the probability that the OS is updated at each step possible. For example, the probability that at time $t$ the OS is still the first release is $(1 - p)^{t-1}$ (i.e., in each of the $t$ time steps the OS was not updated) and the probability that the OS is the most recent release is $p^t$ (i.e., in each of the $t$ time steps the OS was updated). D) However, in this paper we are interested in measuring time in a natural scale, such as days. In that case, we require knowing the most recent OS release as a function of time. We use the schedule of OS updates shown here for the computations that follow.

We first derive an equation for the probability that a component is functional in terms of transitions between the two states. We then develop a metric of performance of the system in terms of the number of functional and nonfunctional components; the former increase performance of the system and the latter decrease performance of the system. We use stochastic simulation to develop intuition about the behavior of an individual component and the properties of a system of many components.

We then derive the Kolmogorov forward (or Fokker–Planck) equation [20, 21] for the distribution of the number of functional components at a future time, starting with a known number of functional components. We use numerical methods to show that the solution of the Kolmogorov forward equation approaches a steady state distribution, which can be anticipated because of general central limit theorems in stochastic population biology [14–16], and derive a Gaussian approximation for the steady state, which requires knowing only the properties of transitions between functional and nonfunctional states for individual components. This Gaussian approximation is both highly accurate and scalable to systems with large numbers of cyber components.

## Systems with multiple possible states

In this case, we focus on the common situation in which the OS of the cyber components is updated as time goes forward and cyber components do not return to a previous OS after updating. In Fig 2A, we show two representations of the OS in use (y-axis) in a subset of about 335 computers from a network of about 7000 computers between July 2019 and March 2020 (details on how the data were chosen can be found in S1 Appendix in S1 File) as a function of the most recent OS release (x-axis). The x-axis measures time in OS release number, and the y-axis indicates which OS is in use at each time. In this case, the most recent OS is predominantly used, as would be common in a business or technology company. In other cases, for example consumers with home computers, a similar plot might show a greater representation of older OS releases.

Furthermore, we will assume that different OS releases have different levels of resistance to exploitation by malware, which is a key factor in determining whether cyberattacks are successful [23, 24]. To model systems with multiple OS releases, in addition to the time course of OS releases, we must specify a model for the process of updating OSs when a new OS is released—because not every cyber component will be updated to the newest OS at the time of its release—and a model for compromise by malware of different OS releases.

We use stochastic simulation to capture the pattern of the data shown in Fig 2A, derive the forward equation for the probability that at a given time a cyber component is using a given OS and that the OS is unexploited, then solve the forward equation numerically. We thus predict, conditioned on the time schedule of the release of OSs and the vulnerability of each OS to malware, in a system with $N_0$ cyber components for which the initial distribution of OSs is specified the subsequent distribution of OSs and the distribution of uncompromised OSs.

## Characterizing time

We briefly address the somewhat philosophical issue of whether models of cyber systems are better using continuous or discrete representations of time. In some cases, there is a clear reason to choose one representation of time over the other. For example, in Fig 2A, the time interval on the x-axis is discrete because each time step corresponds to an update in OS. On the other hand, time on the x-axis in Fig 2D could be either discrete or continuous.

We choose to work in discrete time because it makes much of the mathematical analysis simpler (for examples using continuous time models see [2, 7, 11, 12, 19]). Furthermore, for simplicity, we use the word "days" to describe time, but this is for ease of verbal communication.

Discrete and continuous time can be connected as follows. In discrete time, one specifies the probability, generically denoted by $p$, of a transition of some kind during the next interval of time $\Delta t$; $\Delta t = 1$ in most discrete time models (as with ours). In continuous time, one specifies the rate, generically denoted by $\lambda$, of the transition. The connection between the two is then $p = 1 - e^{-\lambda \Delta t}$ [7]. Note that when $\Delta t = 1$, this becomes $p = 1 - e^{-\lambda}$, so that if the transition probability is specified, the implied rate (with units of per day) is $\lambda = -log(1 - p)$. In the limit of $\Delta t \rightarrow 0$, $p = \lambda \Delta t + o(\Delta t)$, where $o(\Delta t)$ corresponds to terms that are higher powers in the time increment, so that the probability of a transition in the next $\Delta t$ units of time is approximately $\lambda \Delta t$. More details can be found in S2 Appendix in S1 File where we explicitly derive the continuous time analogue of Eqs 3–5 below. In addition, when we derive other key equations, we indicate what changes would occur if the model was in continuous time.

## Methods

We treat systems with only two possible states or those with regular updating of the maximum value of the OS separately. For the former, we first derive the dynamics that a single cyber component is functional, then introduce a metric of performance for a system with $N_0$ components. We then derive the Kolmogorov forward equation for the number of functional components, solve it numerically, and develop a Gaussian approximation for the quasi-steady state solution of the forward equation.

For systems in which the OS is regularly updated, we use the trajectory of the OS shown in Fig 2D and model the distribution that characterizes transitions between a given OS and a more recent OS. We then use stochastic simulation to recover the pattern shown in Fig 2A and explore how parameters of the model cause the pattern to vary. After that, we derive for the forward equation and include compromise of OSs by malware to compute the probability that a cyber component has a specified OS release that is functional at a given time.

### Systems with only two possible states

In a cyber system consisting of $N_0$ components with only two possible states, at each time $t$ each cyber component can be either functional (F) or nonfunctional (NF). At times we simplify notation and describe functional states as "up ($u$)" and nonfunctional ones as "down ($d$)" (as in "my computer is down"). Our first goals are to i) model the dynamics of compromise of individual components and ii) link the dynamics of an individual component to the performance of the entire system.

We let $P_F(t)$ denote the probability that an individual cyber component is functional at time $t$, so that $P_{NF}(t) = 1 - P_F(t)$ is the probability that it is nonfunctional. To be functional at time $t + 1$, a component either had to be functional at time $t$ and remain that way or nonfunctional at time $t$ and transition from a nonfunctional to functional state. We introduce the transition probabilities:

$$p_{uu} = \Pr\{\text{functional component remains so the next day}\} \tag{1}$$

$$p_{du} = \Pr\{\text{nonfunctional component is functional the next day}\}. \tag{2}$$

We interpret $p_{uu}$ as the reliability of the cyber component and $p_{du}$ as the maintainability of the cyber component (*sensu* Mangel [25]). The complement of these transition probabilities is $p_{ud} = 1 - p_{uu}$, which is the probability that a component that is functional now is nonfunctional at the next time period, and $p_{dd} = 1 - p_{du}$, which is the probability that a component that is nonfunctional now is also nonfunctional at the next time period.

To be functional at time $t + 1$, a component must either be functional at time $t$ and remain so or nonfunctional at time $t$ and transition to functional. Thus, the probability that a component is functional at time $t + 1$ satisfies the balance equation

$$P_F(t + 1) = P_F(t)p_{uu} + P_{NF}(t)p_{du} = P_F(t)p_{uu} + (1 - P_F(t))p_{du}. \tag{3}$$

The steady state probability that an individual component is functional is found by setting $P_F(t + 1) = P_F(t) = \bar{P}_F$, from which we obtain

$$\bar{P}_F(1 - p_{uu} + p_{du}) = p_{du}. \tag{4}$$

Since $1 - p_{uu} = p_{ud}$,

$$\bar{P}_F = \frac{p_{du}}{p_{ud} + p_{du}}. \tag{5}$$

Note that if $p_{du}$ is proportional to $p_{ud}$, we obtain the same value of long-term probability of the component being functional regardless of the particular values of $p_{du}$ and $p_{ud}$. That is, if we write $p_{du} = \kappa p_{ud}$, we conclude $\bar{P}_F = \frac{\kappa}{1+\kappa}$. In this case, $\kappa$ determines the rate at which the steady state is approached (see Results) but not the value of the steady state.

In S2 Appendix in S1 File, we derive the continuous time analogues of Eqs 3–5.

**A metric of performance for a system with $N_0$ cyber components.** In general, the metric of performance will be specific to the purpose of the particular cyber system and any physical system that it enables [6, 26–28], but it is also possible to consider a generic metric of performance determined by the state of the cyber system [6].

We assume that the performance of the cyber system, on a scale of 0 to 1, when there are $n$ functional components and $N_0 - n$ nonfunctional components is the product of two sigmoid curves (more details in Mangel and McEver [6]),

$$\varphi(n) = \left( \frac{1}{1 + e^{\frac{n_{50} - n}{\sigma_n}}} \right) \cdot \left( \frac{1}{1 + e^{\frac{(N_0 - n) - n'_{50}}{\sigma'_n}}} \right), \tag{6}$$

where $n_{50}$ is the number of functional cyber components at which the first sigmoid on the right-hand side of Eq 6 is 1/2, $\sigma_n$ is a shape parameter (smaller values of $\sigma_n$ correspond to a sharper transition of the sigmoid values close to 0 to values close to 1), $n'_{50}$ is the number of nonfunctional cyber components at which the second sigmoid on the right-hand side of Eq 6 is 1/2, and $\sigma'_n$ is a shape parameter with a role similar to $\sigma_n$ (Fig 3).

Eq 6 can accommodate a variety of assumptions about the performance of the cyber system. For example, if performance is approximately a linear function of the number of functional components, a larger value of $\sigma_n$ is appropriate. On the other hand, if a communications system requires only one or just a few functional components for a message to get through, small values of $n_{50}$ and $\sigma_n$ are appropriate. Similarly, if the system fails to perform only if many components are nonfunctional, $n'_{50}$ can be set to a moderately large value. And alternately, if the system fails with just a few nonfunctional components, $n'_{50}$ will be small. One can thus choose the parameters based on the particular system. Mapping the specifics of a system onto the sigmoidal functions requires detailed knowledge of that particular system, which is beyond the scope of this paper.

**Dynamics of the system via stochastic simulation.** To develop intuition about the dynamics of the system using stochastic simulation, we specify the transition probabilities $p_{ud}$ and $p_{du}$, the parameters of the performance function, the number $N_0$ of components, and a time horizon $T$ for the simulation, and simulate the transition process $S$ times. We let $\tilde{N}_s(t)$ denote the number of functional components in the $s^{th}$ simulation at time $t$.

If we assume that components make transitions independently, then, given $\tilde{N}_s(t)$, the number of nonfunctional to functional transitions in a day follows a binomial distribution with parameters $N_0 - \tilde{N}_s(t)$ and $p_{du}$. Letting $\mathcal{B}(N, p)$ denote a binomial distribution with

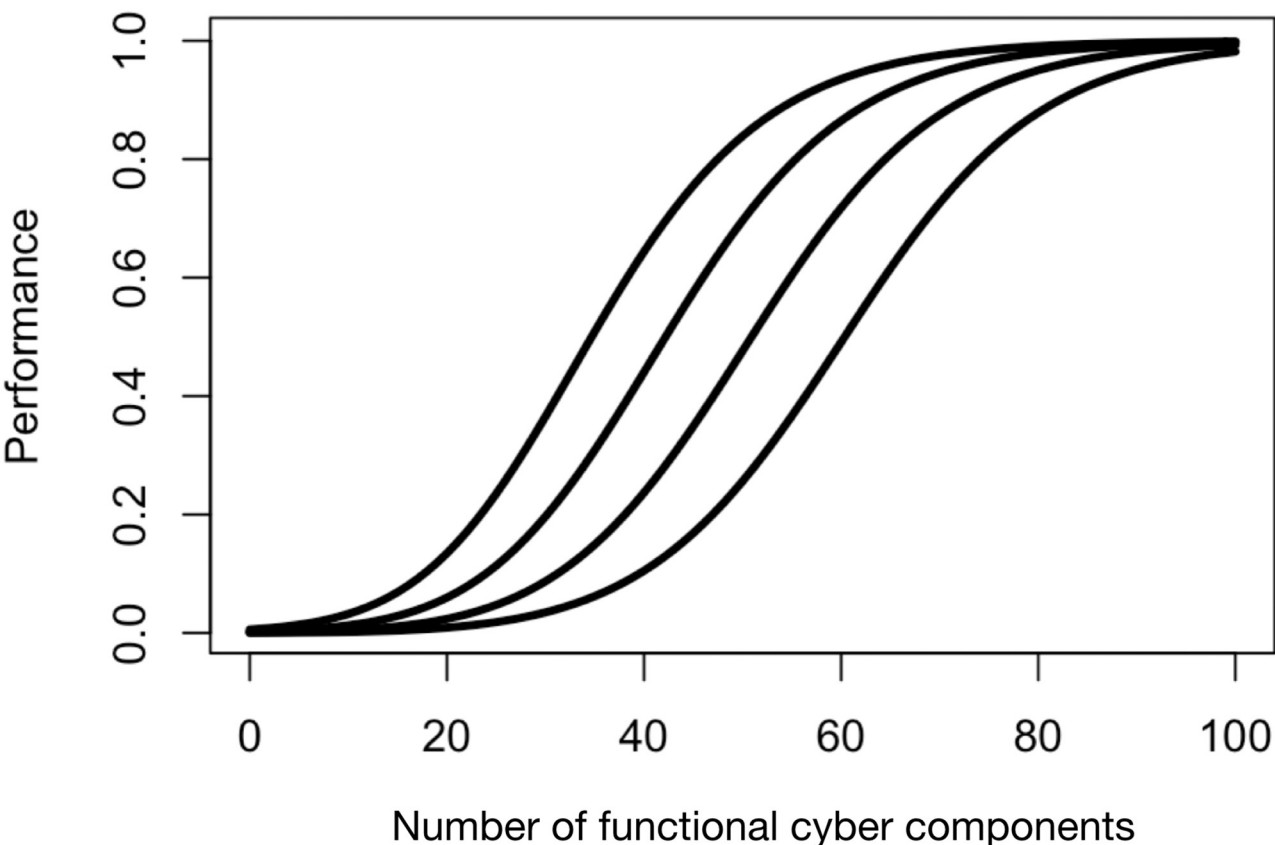

**Fig 3. The double-sigmoid performance function, Eq 6.** In all cases, $n_{50} = 20$, $\sigma_n = \sigma'_n = 10$, and $n'_{50}$ varies from 70 (left-most curve) to 40 (right-most curve). The consequence of increasing $n'_{50}$ is that more components must be functional to achieve the same level of performance.

parameters $N$ and $p$, we write

$$\Delta N_+ \sim \mathcal{B}(N_0 - \tilde{N}_s(t), p_{du}). \qquad (7)$$

Similarly the number of functional to nonfunctional transitions is binomially distributed with parameters $\tilde{N}_s(t)$ and $p_{ud}$,

$$\Delta N_- \sim \mathcal{B}(\tilde{N}_s(t), p_{ud}). \qquad (8)$$

The dynamics of $\tilde{N}_s(t)$ are then

$$\tilde{N}_s(t+1) = \tilde{N}_s(t) + \Delta N_+ - \Delta N_-. \qquad (9)$$

In Discussion, we address the binomial assumption and alternatives to it.

**The Kolmogorov forward equation for the number of functional components.** The Kolmogorov forward equation [7, 11, 20] characterizing variability of the system of $N_0$ components is an analytical approach rather than a simulation. In general, the forward equation requires numerical solution, but unlike with a simulation, one always obtains the same result given the same initial conditions. In physics and engineering, the Kolmogorov forward equation is sometimes known as the Fokker–Planck equation [29, 30].

We let $N(t)$ denote the number of functional cyber components at time $t$. Assuming that $\tilde{N}(0)$ is given, we seek

$$F(n, t) = Pr\{\tilde{N}(t) = n | \tilde{N}(0)\}. \tag{10}$$

If all components are initially functional, $F(N_0, 0) = 1$ and $F(n, 0) = 0$ for any $n \neq N_0$. In general, the only constraint on the distribution for the number of functional components at $t = 0$ is that $\sum_{n=0}^{N_0} F(n, 0) = 1$.

The equation that $F(n, t)$ satisfies can be derived in a relatively straightforward way. We wish to know the probability that the number of functional cyber components at time $t$, $N(t)$, equals $n$. Imagine that at time $t - 1$, the number of functional components is $N(t) = m$ so that there are $N_0 - m$ nonfunctional components at time $t - 1$. If $j$ of the nonfunctional components make the transition NF → F, which has the binomial distribution $\mathcal{B}(j | N_0 - m, p_{du})$, and $k$ of the functional components make the transition F → NF, which has the binomial distribution $\mathcal{B}(k | m, p_{ud})$, then at the next time step, the number of functional components will be $m + j - k$. Whenever $m + j - k = n$, the triple $m, j, k$ will lead to $N(t) = n$ from $N(t) = m$. Thus, if we let $\mathcal{I}_{n;m,j,k}$ denote an indicator function that is 1 if $n = m + j - k$ and 0 otherwise, $F(n, t)$ satisfies

$$F(n, t + 1) = \sum_{k=0}^{m} \sum_{j=0}^{N_0 - m} \mathcal{B}(k | m, p_{ud}) \mathcal{B}(j | N_0 - m, p_{du}) \mathcal{I}_{n;m,j,k} F(m, t). \tag{11}$$

In [2] similar reasoning is applied to a model of networks on a chip in continuous time. Eq 11 is an example of a master equation characterizing the dynamics of probabilities and probability densities [13, 31–34]. In continuous time, the analogue of Eq 11 is a partial differential (in time)-difference (in component numbers) equation. In general, such equations require numerical solution, so we choose to work with the discrete time formulation from the outset.

**Approximation of the quasi-steady state of the forward equation.** In general, the solution of Eq 11 approaches a quasi-steady state as $t$ increases, in which there are only small differences between $F(n, t)$ and $F(n, t + 1)$; we denote this quasi-steady state distribution by $\bar{F}(n)$. Based on ideas about central limit theorems in probability theory [14–16, 20, Ch 7], we seek a Gaussian approximation to the quasi-steady state solution of Eq 11.

Because the number of components takes integer values, we use a discrete Gaussian density. That is, if $m_s$ and $v_s$ denote the approximate steady state mean and variance of $\bar{F}$, we assume that

$$\bar{F}(n) = c_{norm} exp\left[-\frac{(n - m_s)^2}{2v_s}\right], \tag{12}$$

where $c_{norm}$ is a normalization constant chosen so that $\bar{F}(n)$ normalizes to 1,

$$c_{norm} = \frac{1}{\sum_{n=0}^{N_0} exp\left[-\dfrac{(n - m_s)^2}{2v_s}\right]} \tag{13}$$

In S3 Appendix in S1 File, we show that

$$m_s = N_0 \cdot \frac{p_{du}}{p_{ud} + p_{du}},$$

(14)

$$v_s = \frac{(N_0 - m_s)p_{du}(1 - p_{du}) + m_s p_{ud}(1 - p_{ud})}{1 - (1 - p_{du} - p_{ud})^2}.$$

(15)

Comparison with Eq 5 shows that $m_s = N_0 \cdot \bar{P}_F$.

As will be seen in the Results, this Gaussian approximation is very accurate and, unlike the numerical solution of the forward equation, can be scaled up to a large number of components without a great increase in computational time.

## Systems with many possible states

We now consider the situation in which each component can be in one of many possible states. As described above, our motivation is that each cyber component can have OS 1 (the initial release), 2 (the first update), and so forth up to the most recent OS, which we denote by $K(t)$ at time $t$. In general, OS releases will be the most current for varying amounts of time. For this reason, the trajectory $K(t)$ has to be specified; we use the trajectory in Fig 2D for computations.

In addition, we assume that OSs are continually subject to malware attacks. When such an attack is successful, the cyber component becomes nonfunctional. We focus on determining the probability $p(k, t)$ that at time $t$ a cyber component has $OS(t) = k$ and the OS is functional. In the Discussion, we explain how the status of a system with many cyber components can be assessed knowing the properties of a single cyber component.

The stochastic process associated with updating operating systems is a time-dependent, right-limited, pure birth process [35, 36]. It is time-dependent (*sensu* Waugh [37]) because the range of possible transitions changes as time goes forward. It is right limited because at time $t$, the maximum value of $OS(t) = K(t)$, which changes as $t$ increases. It is a pure birth process because of the assumption that the value of $OS(t + 1)$ can only be equal to or greater than $OS(t)$. We will assume, for simplicity, a new release of the operating system occurs between the discrete time steps.

Since the most recent OS release at time $t$ is $K(t)$, the OS for a cyber component ranges $k = 1, 2, \ldots K(t)$. We first characterize the process by which OSs are updated, then simulate this stochastic process to capture the pattern shown in Fig 2A. We then derive the appropriate forward equation and use the solution of the forward equation and a model of exploitation to characterize the probability that an individual component has a functional OS $k$ at time $t$.

**The OS updating process.** We let $l$ denote the OS at time $t$ so that $l = 1, 2, \ldots K(t)$. Since no update can occur until $K(t) > 1$, the OS is equal to 1 until the first time for which $K(t) > 1$. Similarly, if $l = K(t)$, no update can occur.

Otherwise, we assume that the probability that the OS does not change between time $t$ and time $t + 1$ is $e^{-\theta}$, where $\theta$ is the rate at which OSs are updated. (In a continuous time model, the corresponding probability of an update in the next $\Delta t$ units of time is $1 - e^{-\theta \Delta t}$; see S2 Appendix in S1 File). When a transition does not occur, the OS at time $t + 1$ is also $l$.

The probability that the OS is updated is $1 - e^{-\theta}$. We let $f(l, k, t)$ denote the probability of transitioning from OS = $l$ to OS = $k$ given that an update in OS occurs. Since we condition on a jump in OS, given that $K(t) = k$, the range of $l$ is $1, 2, \ldots k - 1$ and the normalization condition that $\sum_{l=1}^{k-1} f(l, k, t) = 1$.

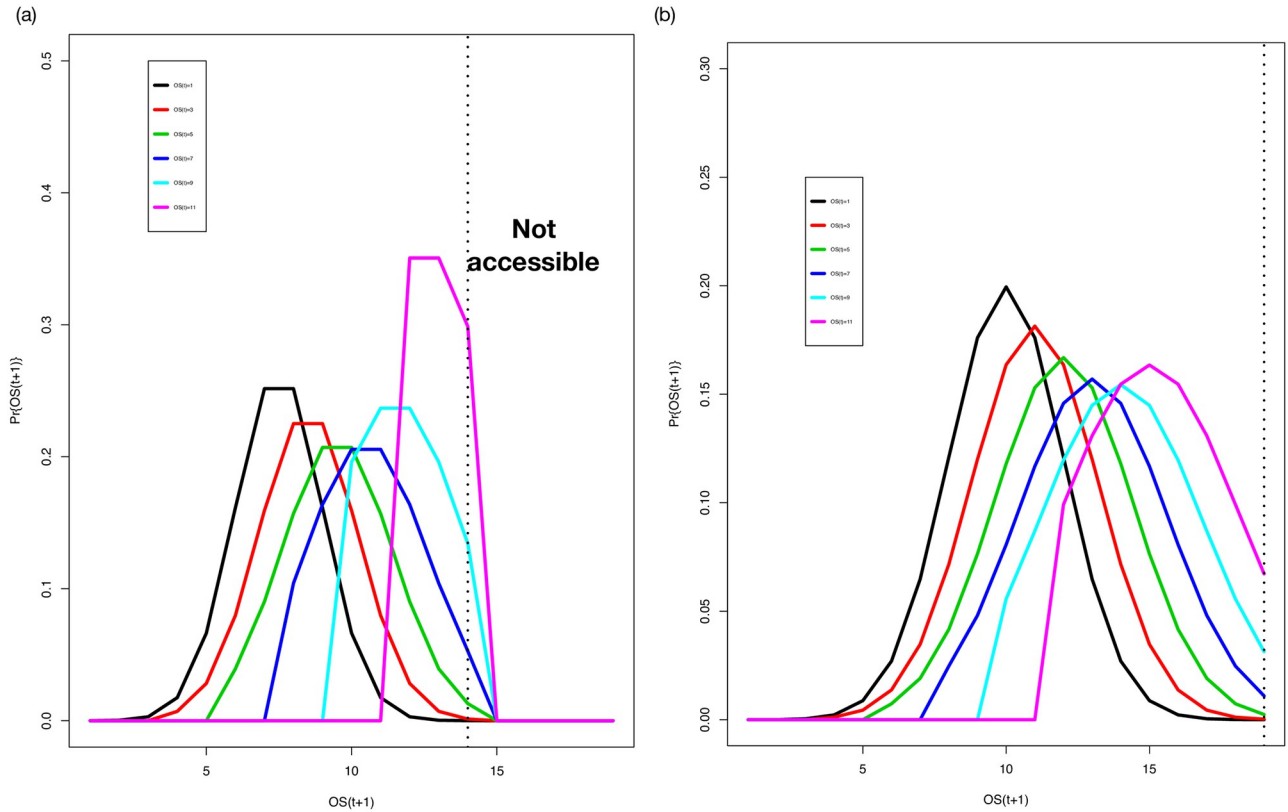

**Fig 4. The updating distribution $f(OS(t), OS(t + 1), t)$ for $t = 190$ (left panel) or 290 (right panel) and six choices of $OS(t)$: 1 (black), 3 (red), 5 (green), 7 (dark blue), 9 (light blue), and 11 (pink).** The dotted lines show regions that are not accessible for updating at that time (determined by the trajectory of the most recent OS, as in Fig 2D). For example since $K(190) = 15$, at time $t = 190$ any OS update beyond 15 corresponds to a time in the future.

For computation, we assume that $f(l, k, t)$ is a discrete Gaussian distribution with a mean-like parameter $\mu(l, t)$ and variance-like parameter $\sigma(l, t)^2$,

$$f(l, k, t) = \frac{exp(-(k - \mu(l, t))^2 / 2\sigma(l, t)^2)}{\sum_{k'=l+1}^{K(t)} exp(-(k' - \mu(l, t))^2 / 2\sigma(l, t)^2)}, \tag{16}$$

which affords great flexibility through the choices of $\mu(l, t)$ and $\sigma(l, t)$. For illustrative computations, we choose $\mu(l, t) = (l + K(t))/2$ and $\sigma(l, t) = 0.2\mu(l, t)$ so that the coefficient of variation of the distribution of jumps is 20%. An alternative is that the variance parameter is constant, then the coefficient of variation would be state and time dependent. Comparison with data will allow one to determine from data which assumption has higher fidelity to operational situations. In Fig 4, we show the updating distributions for a variety of values for $l$ at $t = 190$ and $t = 290$. Eq 16 is sufficiently general that it can accommodate most empirical distributions for updating.

**Simulation to capture the pattern of Fig 1.** We capture the pattern shown in Fig 2A using a stochastic simulation with the following pseudocode:

- Specify the rate of updating $\theta$, the number $N_0$ of operating systems simulated, and the trajectory of operating systems $K(t)$ for $t = 1, 2, \ldots$ to the end time $T$.

- Let $OS(n, t)$ denote the number of operating systems with release $n$ at time $t$ so that $\sum_{n=1}^{K(t)} OS(n, t) = N_0$.

- For all $n$ set $OS(n, 1) = 1$.

- Cycle over $n$.

- Within the cycle over $n$ for $t = 1$ to $t = T - 1$:

  - If $OS(n, t) = K(t)$, no update is possible; set $OS(n, t + 1) = K(t)$.

  - If $OS(n, t) < K(t)$, draw a random variable $\tilde{Z}$ uniformly distributed on [0, 1]. If $\tilde{Z} > 1 - e^{-\theta}$, the OS is not updated, so $OS(n, t + 1) = OS(n, t)$.

  - If $\tilde{Z} \leq 1 - e^{-\theta}$, apply the updating distribution $f(OS(n, t), k, t)$ from Eq 16. Draw a random variable $\tilde{Z}_1$ uniformly distributed on [0, 1] and find $k$ satisfying $\sum_{j=OS(n,t)+1}^{k} f(OS(n, t), j, t) < \tilde{Z}_1 \leq \sum_{j=OS(n,t)+1}^{k+1} f(OS(n, t), j, t)$. Set $OS(n, t + 1) = k$ and continue cycling over time.

**The forward equation.** To derive the forward equation for the probability $p(k, t)$ that at time $t$ an individual component has operating system $k$, we use the method of thinking along sample paths [7], as illustrated in Fig 5.

Clearly, until $K(t) = 2$, $p(1, t) = 1$ and $p(k, t) = 0$ for $k > 1$. Furthermore, at any time $t$, the OS will be the first release if no transition occurred during the entire interval; thus $p(1, t) = e^{-\theta(t-1)}$.

Once $K(t) > 1$, we separate two situations. First, $OS(t + 1) = k$ if $OS(t) = k$ and no updating transition occurred; this event has probability $p(k, t)e^{-\theta}$. Second, $OS(t + 1) = k$ if $OS(t) = l$ and a transition occurred from operating system $l$ to operating system $k$; this event has probability $p(l, t)(1 - e^{-\theta})f(l, k, t)$. Thus,

$$p(k, t + 1) = p(k, t)e^{-\theta} + (1 - e^{-\theta})\sum_{l=1}^{k-1} p(l, t)f(l, k, t), \tag{17}$$

where $k = 2, ..K(t)$. The first term on the right-hand side of Eq 17 corresponds to no update in OS between times $t$ and $t + 1$; the second term corresponds to an update from current OS $l$ to a new OS $k$. Since the cyber component must have some operating system, the normalization condition for Eq 17 is $\sum_{k=1}^{K(t)} p(k, t + 1) = 1$.

The continuous time version of Eq 17 is a differential (in time)-difference (in OS number) equation. In general, like Eq 11, it will require numerical solution. When the number of components $N_0$ in Eq 11 or the number of potential OSs in Eq 17 is large, it is possible to obtain approximate analytical solutions by asymptotic methods [31, 38].

**Including compromise by malware.** We let $t_k$ denote the time at which the $k^{th}$ OS is released and assume that the probability that it is not exploited at time $t > t_k$ is $p_{ne}(k, t) = e^{-\lambda_k(t - t_k)}$ where $\lambda_k$ is the rate of attack by malware. For computations, we assume that $\lambda_k$ decreases as $k$ increases (Fig 6a), i.e., that newer OS are more able to resist exploitation. The probability that an individual cyber component has OS $k$ and is unexploited at time $t$ is then $p(k, t) \cdot p_{ne}(k, t)$.

In S4 Appendix in S1 File, we provide commented code, written as simply as possible, for implementing these methods. All computations were done using R Studio 1.0.143 with underlying R 3.6.1 GUI 1.70 El Capitan build (7684) on an iMac running Mac OS 12.1.

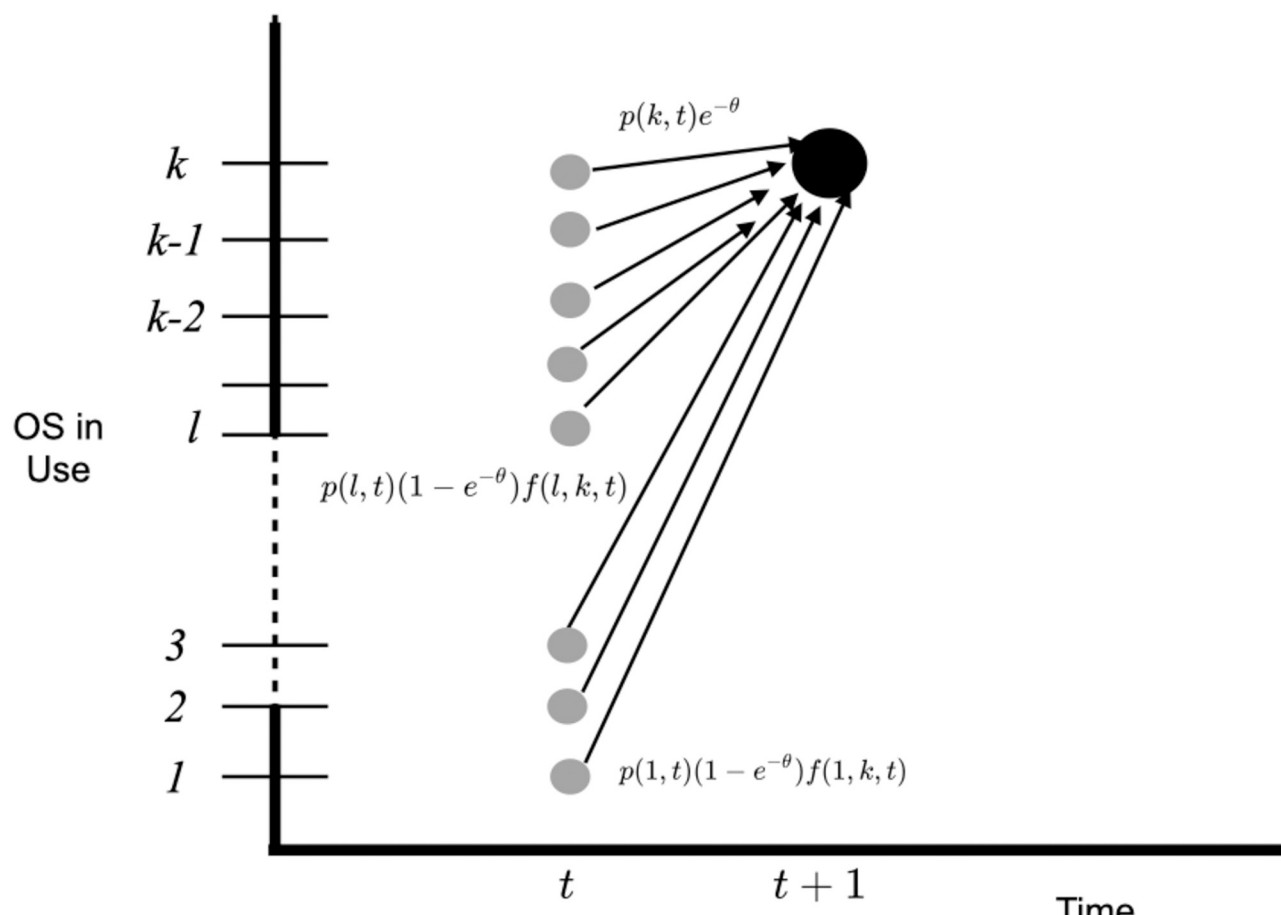

**Fig 5. To derive the forward equation for $p(k, t)$, we use the method of thinking along sample paths.** If $OS(t + 1) = k$, either $OS(t) = k$ and no transition occurred (with probability $e^{-\theta}$), or the $OS(t) = l$, where $l = 1, 2, 3, ..k − 1$, and transition (with probability $1 − e^{-\theta}$) occurred from $OS(t) = l$ to $OS(t + 1) = k$ (probability $f(l, k, t)$).

## Results

We begin with simulation results for systems with only two possible states and show how the number of functional components fluctuates, but that its mean approaches the average predicted from our analysis and leads to a distribution of functional components as a function of time.

We then turn to the solution of the forward equation for systems with only two possible states and show i) examples of the numerical solution of the forward equation for two different initial conditions and ii) the approach to the quasi-steady state and that Gaussian approximation to the quasi-steady state is very accurate.

For systems with multiple possible states, we use stochastic simulation to reproduce the pattern in Fig 2A, exploring how the distribution of OSs in time depends on $\theta$ characterizing the rate at which OSs are updated. We then show examples of the solution of the forward equation, with and without exploitation of the OSs. The right-limited nature of the stochastic process and the effect of exploitation become clear when one considers the solution of the forward equation.

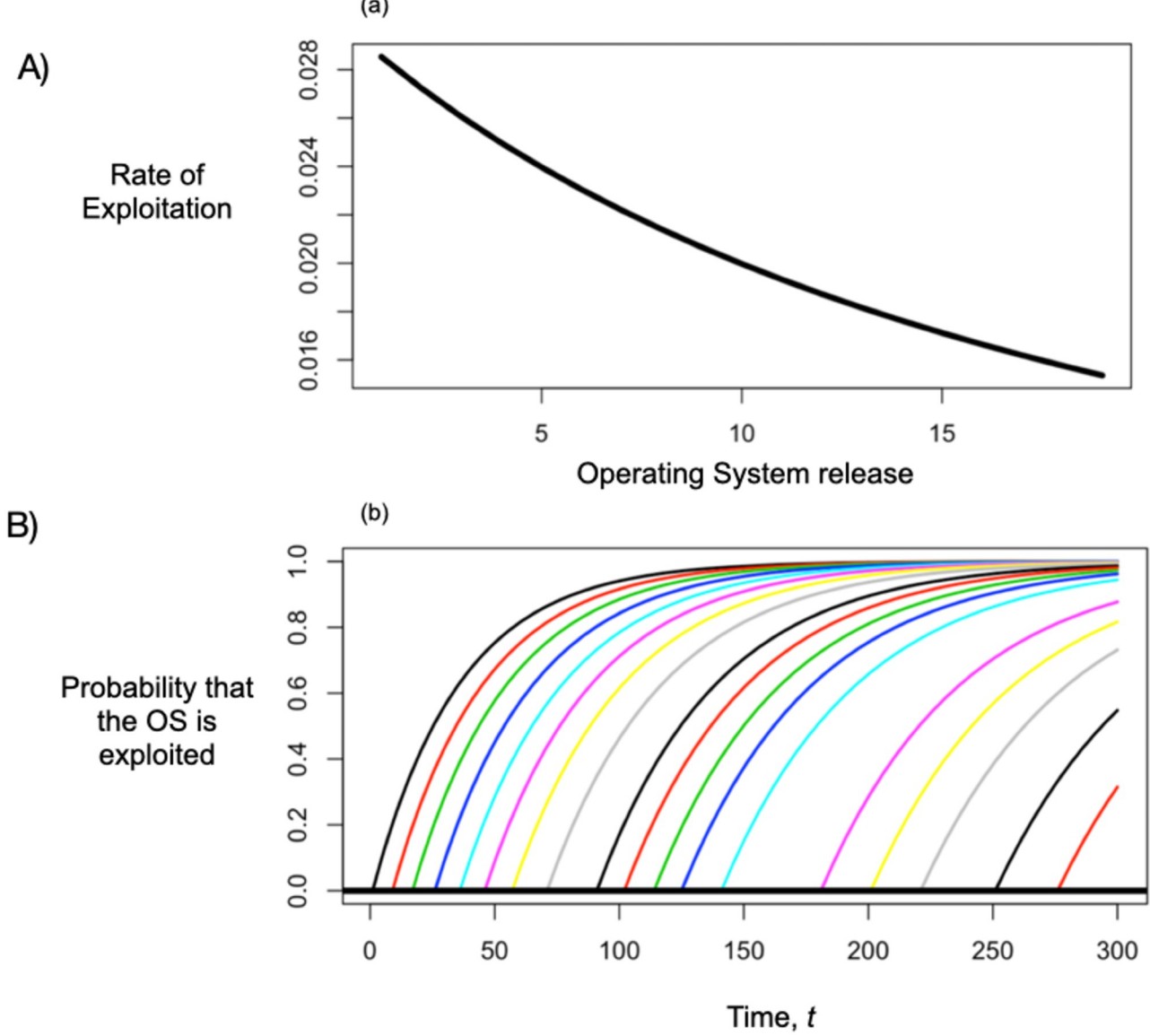

**Fig 6. We assume that the $k^{th}$ OS is released at time $t_k$ and remains unexploited at time $t$ with probability $e^{-\lambda}{}_k(t - t_k)$.** a) For computations, we assume that the rate of exploitation declines as OS release number increases (i.e., newer releases are more capable of resisting exploitation). b) The probability that the OS releases in Fig 2D are exploited as a function of time. By definition, the probability of exploitation is 0 for $t < t_k$; it will approach 1 for $t > > t_k$ because of our assumption of an exponential rate of exploitation.

### Systems with only two possible states

**Intuition via stochastic simulation.** In Fig 7, we show an example of the output of the simulation. It is clear that the mean number of functional components is approaching a value around 71 (as predicted by Eq 14). The simulation allows us to see the variation of individual trajectories around that mean.

Given the transition probabilities for a single component, it is therefore possible to directly predict the mean number of functional components at any time subsequent (see S3 Appendix in S1 File) and the steady state number of functional components (Eq 14). Because the

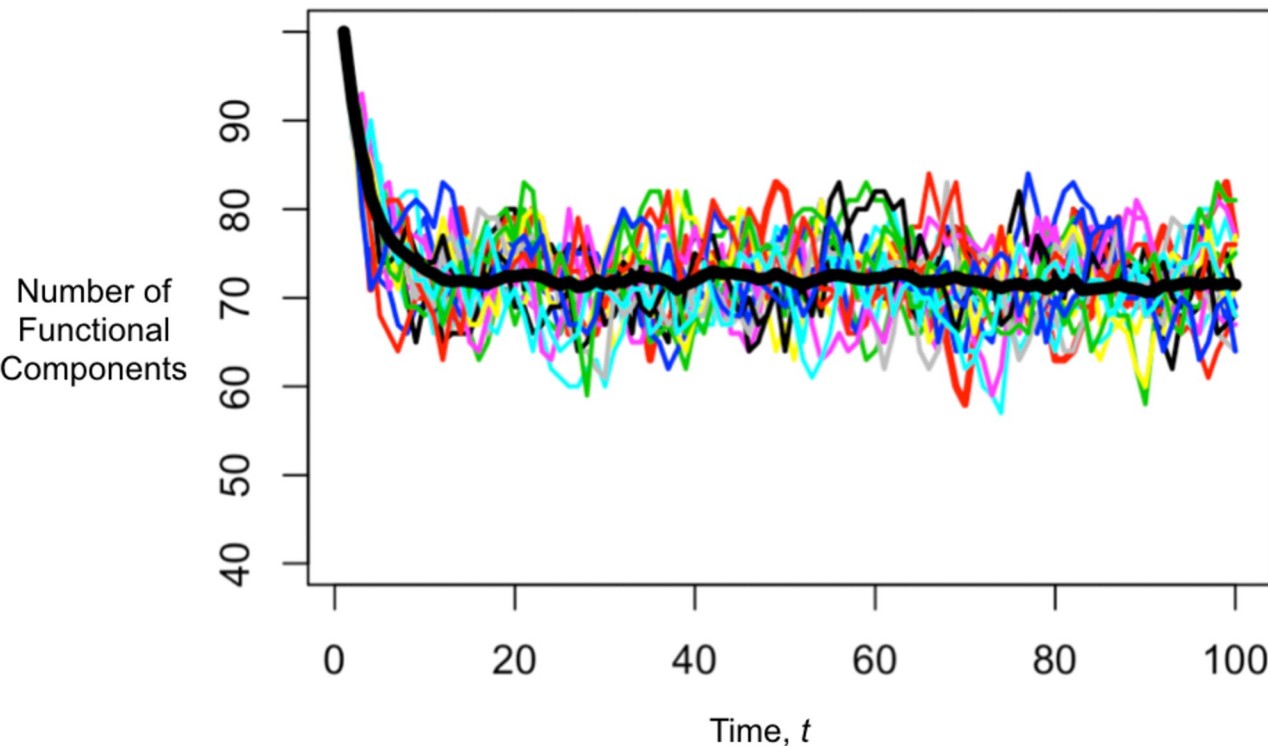

**Fig 7. Fifty simulations of Eq 9 for $N_0 = 100$, $p_{ud} = 0.08$, and $p_{du} = 0.2$.** Here $\bar{P}_U = \frac{p_{du}}{p_{ud}+p_{du}} = 0.714$. The thick black line is the mean value of the individual trajectories.

dynamics follow a linear stochastic difference equation, the trajectory of $\tilde{N}(t)$ will have a mean that declines toward $m_s$ when $\tilde{N}(0) > m_s$ or rises toward $m_s$ when $\tilde{N}(0) < m_s$. Once $\tilde{N}(t)$ is in the vicinity of $m_s$, its trajectory will fluctuate around $m_s$.

We obtain a distribution of performance (Fig 8) by combining Eq 6 for performance with the dynamical processes in Eqs 7–9. Note the approach to a steady state distribution of performance.

**Numerical solution of the Kolmogorov forward equation and the Gaussian approximation.** In Fig 9, we show the numerical solution of the forward equation for two different initial conditions and the same transition probabilities between functional and nonfunctional states. It is clear that $F(n, t)$ approaches a quasi-stationary distribution (the difference in the scale on the y-axis in the two panels somewhat obscure that the means of the quasi-steady states are the same).

In Fig 10, we show the quasi-steady state of the numerical solution of the forward equation and the Gaussian approximation to the quasi-steady state solution for three different sets of transition probabilities between functional and nonfunctional states (left panels) and the Gaussian approximation for the final quasi-steady state (right panels).

In this figure, the reliability of components, measured by $p_{ud}$, is constant, but the ability to repair nonfunctional components, measured by $p_{du}$, declines across rows. Since the overall rate of transitions is lower as $p_{du}$ decreases, it takes longer to reach the quasi-steady state distribution ($t = 20$, 40, or 60 in the three rows). In addition, the peaks of the distributions and the Gaussian approximation shift to the left, consistent with Eq 14.

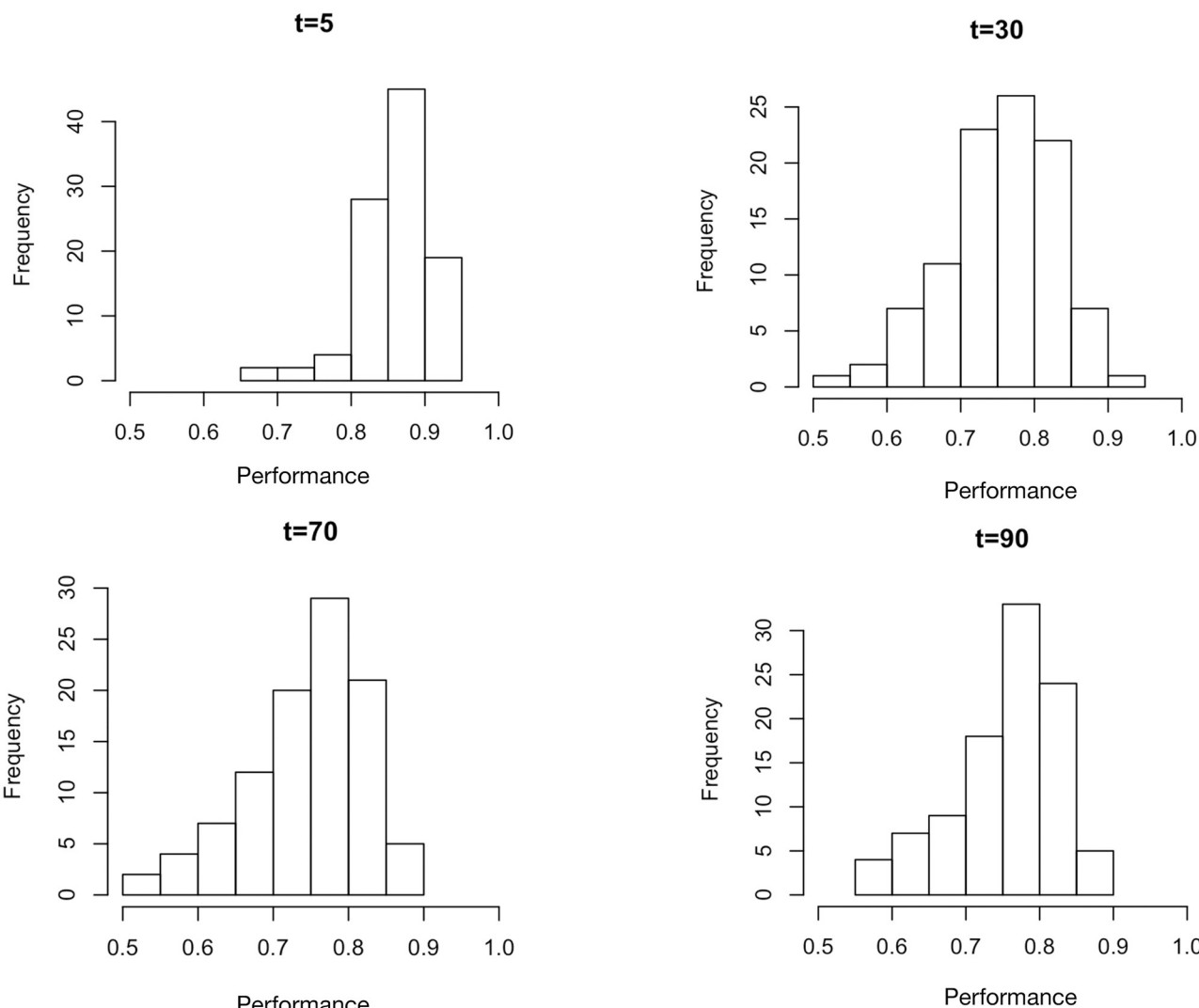

**Fig 8. Frequency of the distribution of performance (Eq 6) at four different times, for $p_{ud}$ = 0.08 and $p_{du}$ = 0.2, with parameters for the performance function $n_{50}$ = 20, $\sigma_n$ = 10, $n'_{50} = 40$, and $\sigma'_n = 10$.** Note the left tail of the distributions expanding as time increases and an approach to a steady state frequency distribution of performance.

It is clear that the discrete Gaussian approximation does very well. In systems with hundreds or even thousands of components, or in cases in which one wants to sweep over ranges of the transition probabilities, the advantage of this approximation over numerical solution of the forward equation or simulation will be very great. A remaining question is how soon after $t = 1$ could the Gaussian approximation be used with $m(t)$ and $v(t)$ as the mean and variance.

## Systems with many possible states

**Recovering the pattern of Fig 1.** We used simulation to recreate the pattern shown in Fig 1 (Fig 11). The distribution of OSs across time, measured by the maximum OS release, varies according to the updating parameter. For $\theta = 0.025$, corresponding to about a 2.5% chance of updating an OS in a day, the distribution of OSs is very dispersed; when $\theta$ increases to 0.05, the dispersion reduces, and when $\theta = 0.1$, we obtain results similar to Fig 2A. Since these are

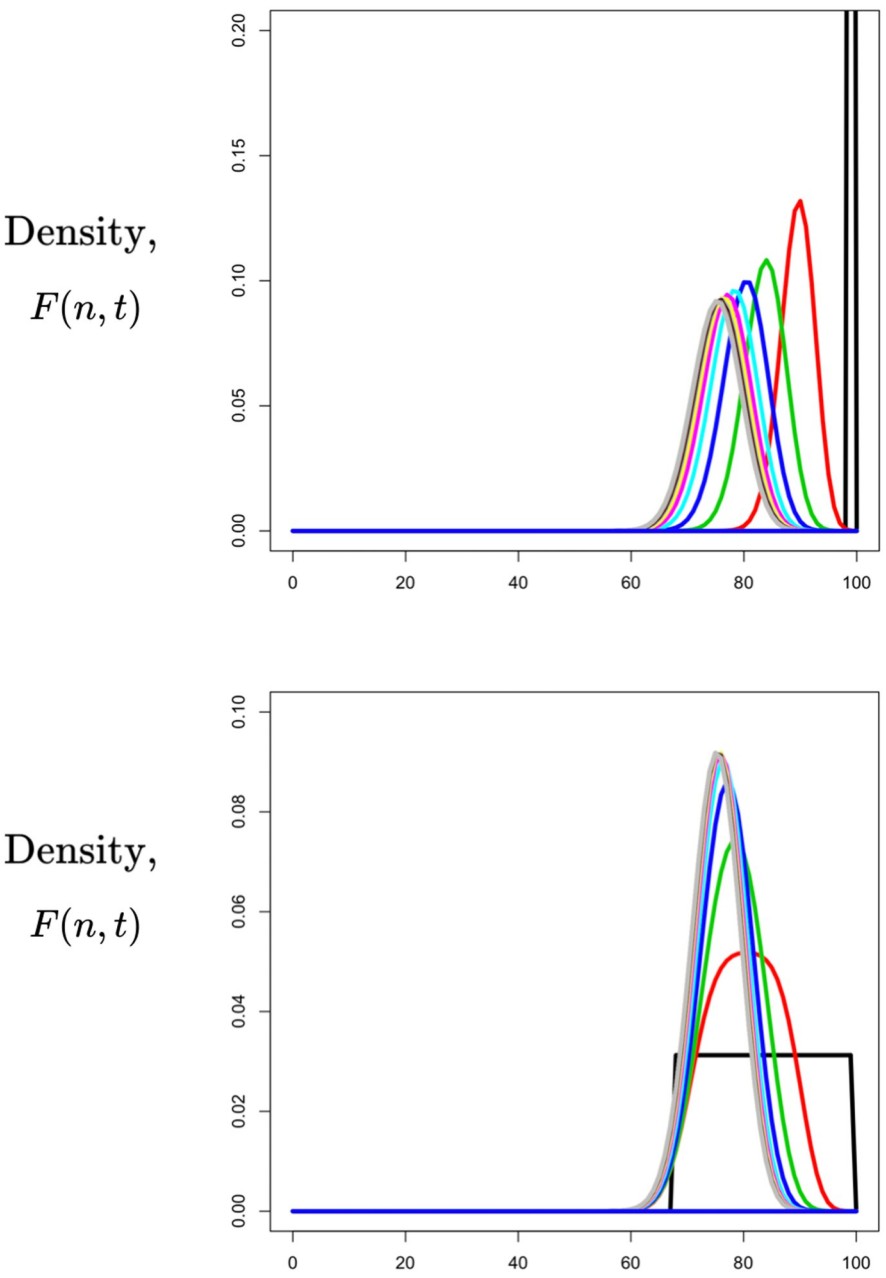

**Fig 9. Numerical solution of the forward equation for $p_{ud}$ = 0.08 and $p_{du}$ = 0.2.** In the upper panel, the initial number of functional components is concentrated at $N_0$ = 100; in the lower panel, it is uniformly distributed between 69 and 100. Each color corresponds to one time step into the future, and time increases as the curves move to the left. Note the different scales in the vertical axes.

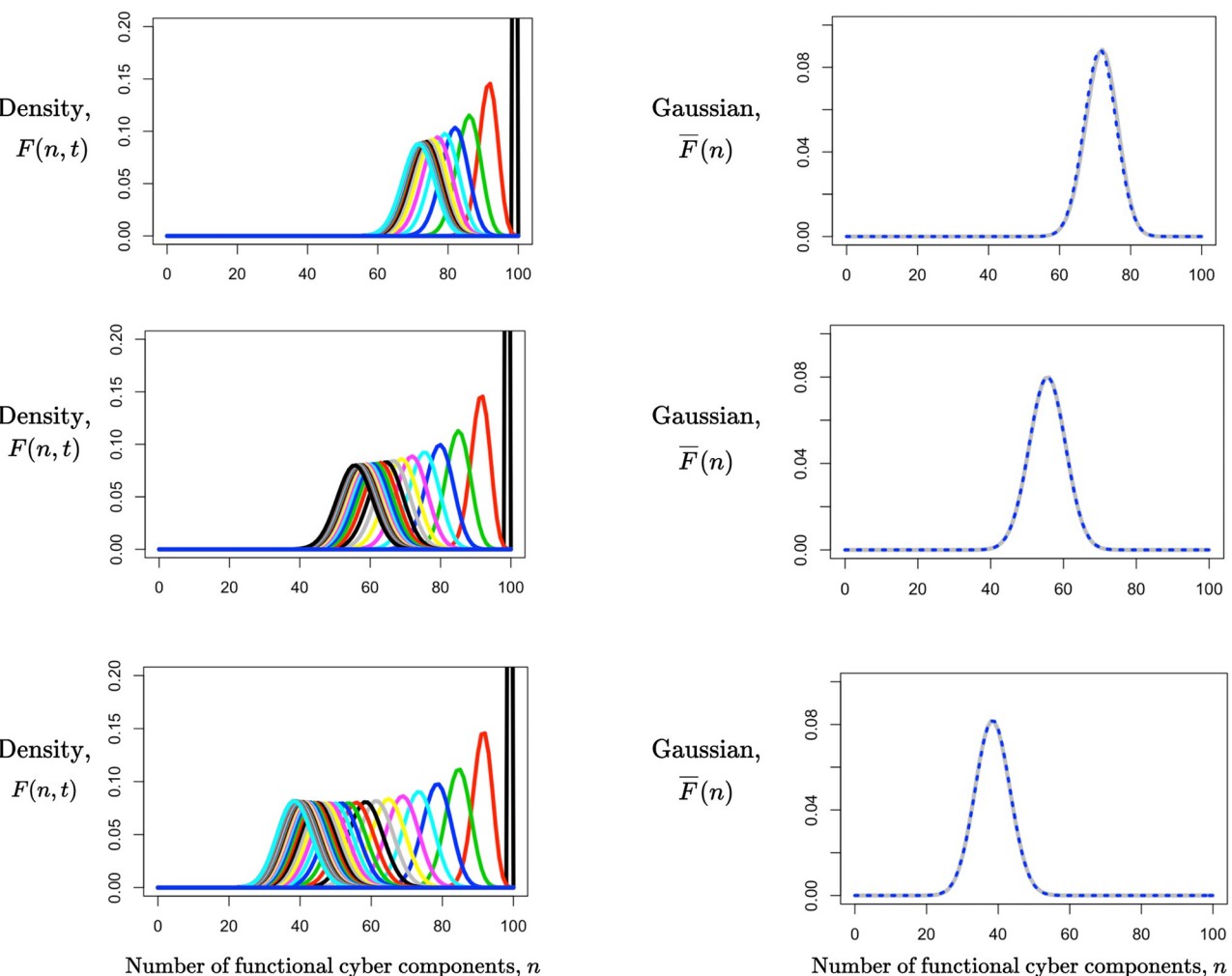

**Fig 10. Comparison of the discrete Gaussian approximation and the quasi-steady state solution of the forward equation for the case of $N_0 = 100$ components, all initially functional.** All x-axes correspond to the number of functional components. In the left-hand panels, we show the numerical solution of the forward equation for $p_{du} = 0.2$ and $p_{ud} = 0.08$ (first row), $p_{du} = 0.1$ and $p_{ud} = 0.08$ (second row), and $p_{du} = 0.05$ and $p_{ud} = 0.08$ (third row), marching toward the quasi-steady state solutions. In the right-hand panels, we compare the discrete Gaussian approximation with the densities at $t = 20, 40,$ and $60$ (upper, lower, and middle rows, respectively) for the same individual component transition probabilities. The solid gray line corresponds to the densities $F(n, 20)$, $F(n, 40)$, and $F(n, 60)$, and the dotted blue line corresponds to the Gaussian approximation.

simulation results, one would expect variation in the quantitative details with additional replications of the simulation but not the qualitative patterns.

**The distribution of unexploited OS in time.** In Fig 12, we show the distribution of OSs over time, determined from the solution of the forward equation (Eq 17) with the updating probability in Eq 16. We see the right-limited nature of the stochastic population process, especially for $t = 90$ and $t = 190$—the distribution is not symmetric but has a long left-hand tail and is necessarily 0 for OS releases that are not available. Comparing the dashed and solid lines, we also see that exploitation makes the frequency distribution of unexploited OSs more symmetrical around its peak because the older an OS release, the more likely it is to be exploited.

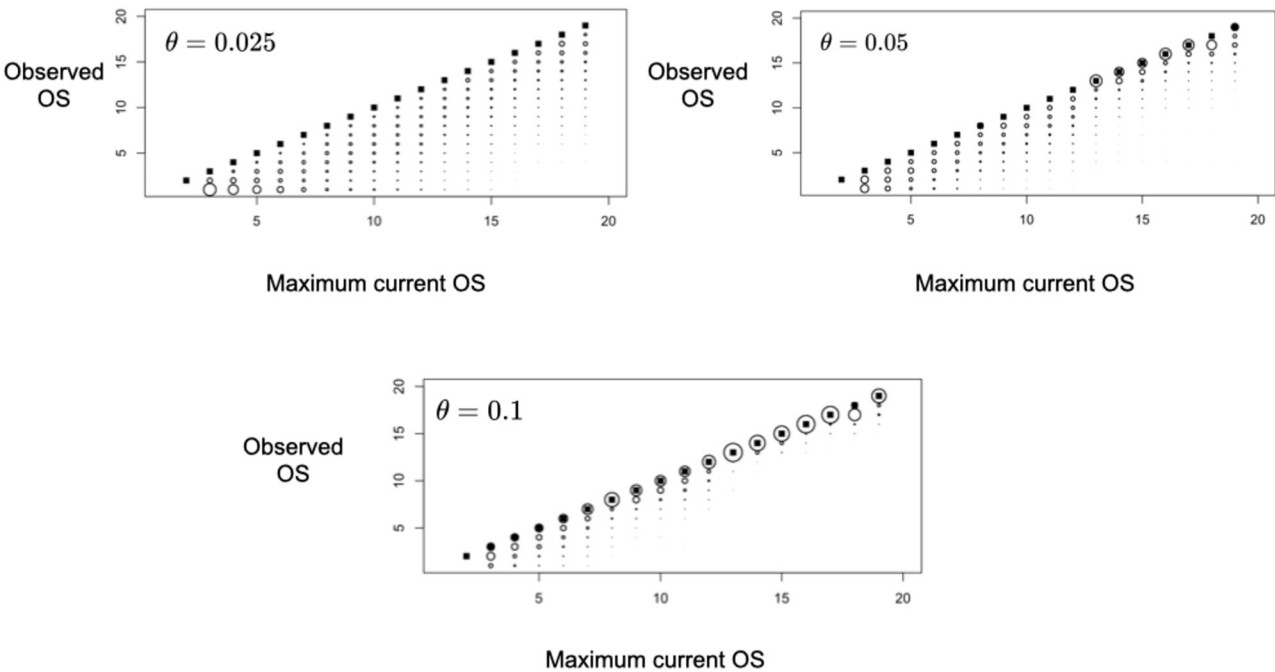

**Fig 11. Simulation of the distribution of 1000 OSs for the updating distribution in Eq 16 and three different values of θ.** Squares show the most recent OS, and circles show the fraction of the 1000 simulated cyber components in the respective OS. For ease of viewing, the fraction is multiplied by 3. This figure should be compared with the data in Fig 2A.

## Discussion

In addition to points raised in the main text and treated in S2 and S3 Appendices of S1 File, our work raises some other general issues.

### Models as a guide for measurement

Both of our models are intended to have much in common with many cyber systems but not as models of a particular cyber system. They therefore serve as a guide for what needs to be measured to apply the ideas to a specific cyber system.

For example, for cyber systems in which the components have only two states, one needs to estimate $p_{uu}$ and $p_{du}$ (as in Eqs 1 and 2), and for a system with multiple states, one needs to determine the trajectory of OS releases (as in Fig 2D) and the parameters of the updating distribution $f(l, k, t)$ in Eq 16. If compromise by malware is included in the study of a system with multiple states, one must also estimate the rate of compromise of different releases of the OS (as in Fig 6).

Similarly, we intentionally chose a very general metric of performance depending on the state of the cyber system to focus on the characterization of cyber variability. As discussed above, the parameters in $\varphi(n)$ can be chosen to specify the mission that the cyber system supports. Furthermore, if one has a particular physical system that depends on the cyber system for its performance, a model of the physical system can be linked to that of the cyber system (see [6] for application to an electric grid).

We recognize that estimating these parameters is not a trivial matter. However, data that are needed to estimate the parameters are already collected by technology staff in many institutions and businesses, and are likely available as proprietary information from major software

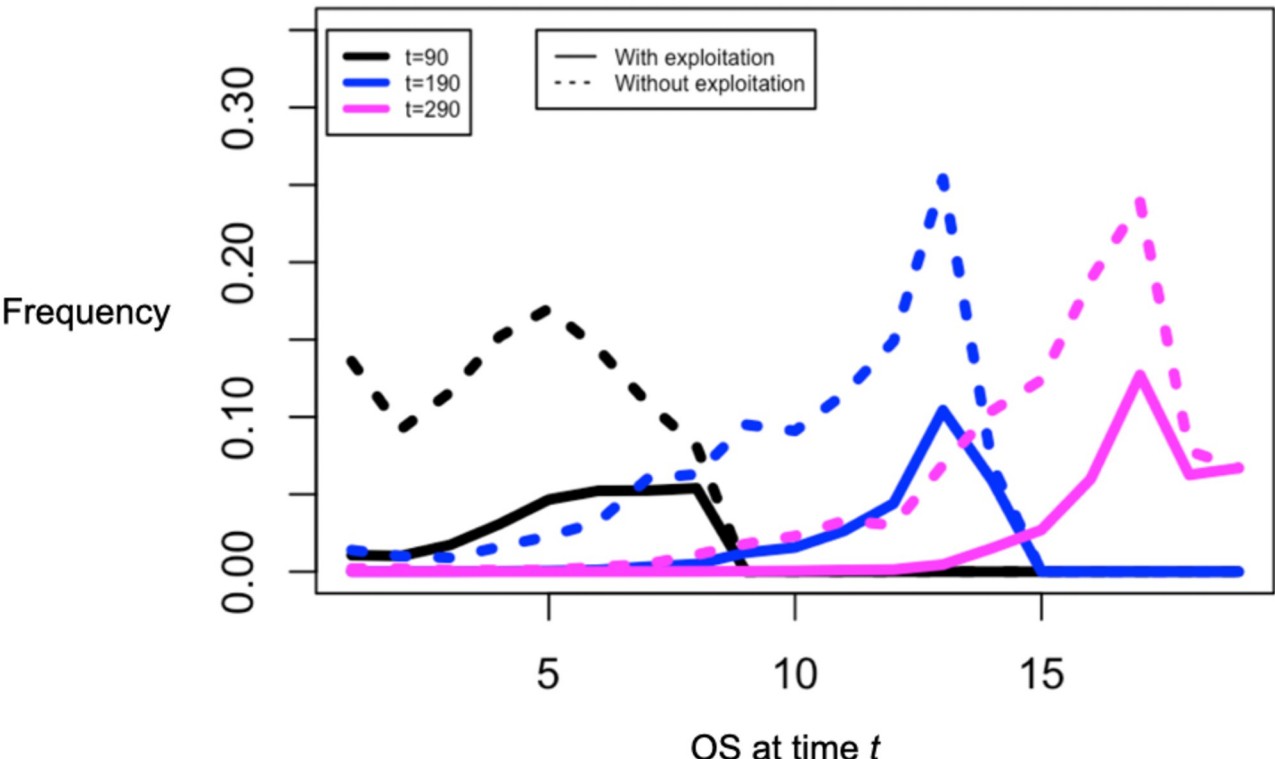

**Fig 12. The distribution of OSs over time for θ = 0.025.** We show the distribution of all OSs as a dotted line—- note the long left-hand—-tail and the distribution of unexploited OSs as a solid line. Because the stochastic process is right limited, the frequency distribution drops to zero for times at which an OS is not available (i.e., is 0 whenever $t < t_k$). In addition, since older OSs are more likely to be exploited than more recent ones, the distribution of unexploited OSs is more symmetrical than the distribution of all OSs.

companies such as Microsoft and Apple. In cases where such data are not collected, our model-ing—as models should do—provides motivation for their collection.

For ease of presentation, we began with cyber systems that can be only functional or non-functional or have many states. Although much can be learned from the model with just two states [25], there is opportunity for an increased, but not too large, level of complexity. For example, a cyber component might be described as fully functional (FF), reduced functional (RF), barely functional (BF), or nonfunctional (NF). Doing so increases the number of transi-tion probabilities considerably so that an analysis similar to the one that we conducted is more daunting. However, determining the parameters in a particular case from operational data is feasible [25].

## Non-binomial transitions

We used binomial transitions between states of the cyber component in the model with two states. It is well known that binomial distribution has a strong central tendency. An alternative would involve aggregated transitions. For example, in malware attacks most days no or few components will fail, but after an attack, many components may fail simultaneously. That is, the number of failures is overdispersed, in the sense that the variance of the number of failures is larger (and sometimes much larger) than the mean. This will lead to overdispersion in com-parison to a model in which the transitions are modeled by a binomial distribution [7].

One way to capture this idea is to give the transition probabilities $p_{ud}$ and $p_{du}$ distributions themselves, which is tantamount to assuming that with each day there is a randomly chosen probability for the F $\rightarrow$ NF transition and an independently randomly chosen probability for the NF $\rightarrow$ F transition.

A natural framework is to give each of the transition probabilities a beta density [39] in which we assign a beta density to the probabilities $p_{ud}$ and $p_{du}$. That is, we assume that $p_{ud}$ has density $\frac{1}{B(\alpha_{ud},\beta_{ud})} p_{ud}^{\alpha_{ud}-1}(1-p_{ud})^{\beta_{ud}-1}$, where $\alpha_{ud}$ and $\beta_{ud}$ are parameters and $B(\alpha_{ud},\beta_{ud})$ is the beta function of those parameters. Given that the number of functional components at a particular time is $n$, the mean and variance of the number of F $\rightarrow$ NF transitions are $\frac{n\alpha_{ud}}{\alpha_{ud}+\beta_{ud}}$ and $\frac{n\alpha_{ud}\beta_{ud}(\alpha_{ud}+\beta_{ud}+n)}{(\alpha_{ud}+\beta_{ud})^2(\alpha_{ud}+\beta_{ud}+1)}$. One would make a similar assumption about the NF $\rightarrow$ F transitions. Although the details of Eqs 14 and 15 will change, the procedure leading to them will not. As long as one can compute the mean and variance of the transition processes, the Gaussian approximation can be developed.

## Non-Markov transition processes

Although non-Markov transition processes are common in the natural world (and thus Markov processes, more an exception than the rule), scientists and engineers tend to seek variables that allow a natural process to be described with Markovian transitions [40, 41]. It is easy to envision (but less easy to account for, and beyond the scope of this paper) non-Markovian transitions in the models we developed.

For example, for the case in which there are only two states, we assumed that the NF to F transition depended only on the number of components that were nonfunctional, i.e., on $N_0 - n$. However, there is likely a distribution of the time needed to repair components that are nonfunctional because of the complexity of the cause making the component nonfunctional. Thus, the probability of a component making the NF to F transition may also depend on how it has been nonfunctional. That is, the NF to F transition may depend on not only how many are components are nonfunctional but also when they arrived. This leads to a composite stochastic process [40] in which nonfunctional components are characterized by both being down and how long they have been down, so that the model for the NF to F transition also requires the probability of transitioning to the functional state as a function of the amount of time a component has been down. An ecological example of this approach can be found in [42] and [43]. The modification of the simulation that generated Figs 7 and 8 is straightforward, but an entirely new Kolmogorov forward equation and associated numerical solution and Gaussian approximation are needed. Doing so is beyond the scope of this paper.

## A system of $N_0$ components with multiple possible states: A combined multinomial and binomial distribution

In a system of $N_0$ components with multiple possible states, we let $\tilde{M}_k(t)$ denote the number of components with OS release $k$ at time $t$ so that $\sum_{k=1}^{K(t)} \tilde{M}_k = N_0$. If updating of OS releases occurs independently, $\tilde{M}_k(t)$ will follow a multinomial distribution so that the expected number and variance of components with OS equal to $k$ are

$$
\begin{aligned}
\mathcal{E}(\tilde{M}_k(t)) &= Np(k,t) \\
Var(\tilde{M}_k(t)) &= Np(k,t)(1-p(k,t)).
\end{aligned}
\tag{18}
$$

and the covariance of OS values is $Cov(\tilde{M}_k(t), \tilde{M}_j(t)) = Np(k,t)p(j,t)$, where $p(k,t)$ is the

solution of Eq 17. Given that $\tilde{M}_k(t) = m_k$, the number of unexploited components with OS $k$ is then binomially distributed with parameters $m_k$ and $p(k, t) \cdot p_{ne}(k, t)$.

## Conclusions

We have shown that treating a cyber system as a metaphorical population in which individual cyber components, much like individual organisms in a biological population, transition between states provides a valuable framework for characterizing variability in cyber systems.

Our first example, in which each individual component can be in only one of two states with transitions between them allowed us to derive a simple dynamic equation for the probability that a component is functional. By introducing a generic performance function depending on the status of all the components, we were able to characterize the frequency distribution of performance of the entire system based on the transition probabilities of individual components.

In this relatively simple system, we numerically solved the Kolmogorov forward equation for the probability that $n$ of the $N_0$ components are functional at time $t$ given the number of functional components at time 0. We showed numerically that the solution of the forward equation approaches a quasi-steady state, then derived a highly accurate Gaussian approximation for the steady state distribution that requires only the transition probabilities for an individual component. This approximation will be valuable in systems with hundreds or thousands (or more) of components in which direct numerical methods are less practicable.

In our second example, we assumed that the cyber system had multiple possible states, as in the updates of the OS for the cyber components. To derive the forward equation in this case, we needed to specify the trajectory of the OS and a transition density that characterized the probability distribution of the update conditioned on the current OS. Doing so allowed us to reproduce the qualitative pattern of OSs in a subset of data from a real-world network of 7000 computers. The solution of the corresponding forward Kolmogorov equation allowed us to predict the distribution of OS in time, and by coupling that with a model for exploitation of OS depending on the time since its release, we were able to compute the distribution of unexploited OS at future times. In this case, as long as OSs continue to be released, there will be no steady state and the frequency distribution of OSs will be dynamic.

## Supporting information

**S1 Fig. The distribution of updates to the Microsoft Windows 10 build 17763 in the network of 7000 computers that we sampled during the first quarter of our data collection.** The colors represent different updates of the build (see text for explanation): 17763.615 (light blue), 17763.557 (yellow), 17763.678 (red), 17763.529 (dark blue), and 17763.503 (gold).
(TIF)

**S1 Table. Details of the data shown in Fig 2A.**
(XLSX)

**S2 Table. Details of the data shown in Fig 2A.**
(XLSX)

**S1 File. This includes a summary of the data shown in Fig 2A, connecting continuous and discrete time models, and the Gaussian approximation for the forward equation.** Codes that implement the methods [44, 45].
(PDF)

## Acknowledgments

We thank Steve Munch for helpful discussions and an anonymous referee for a very helpful review of a previous version of the manuscript.

## Author Contributions

**Conceptualization:** Marc Mangel, Alan Brown.

**Data curation:** Marc Mangel, Alan Brown.

**Formal analysis:** Marc Mangel, Alan Brown.

**Methodology:** Marc Mangel.

**Writing – original draft:** Marc Mangel.

**Writing – review & editing:** Marc Mangel, Alan Brown.

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
