## [Decision Letter · Decision Letter 0]

16 Aug 2022

PONE-D-22-10441Population Processes in Cyber System VariabilityPLOS ONE

Dear Dr. Mangel,

Thank you for submitting your manuscript to PLOS ONE. After careful consideration, we feel that it has merit but does not fully meet PLOS ONE’s publication criteria as it currently stands. Therefore, we invite you to submit a revised version of the manuscript that addresses the points raised during the review process.

Please note that we have only been able to secure a single reviewer to assess your manuscript. We are issuing a decision on your manuscript at this point to prevent further delays in the evaluation of your manuscript. Please be aware that the editor who handles your revised manuscript might find it necessary to invite additional reviewers to assess this work once the revised manuscript is submitted. However, we will aim to proceed on the basis of this single review if possible. The reviewer has raised a number of queries and suggestions that need to be carefully addressed in your revisions; please refer to their detailed comments below.

We look forward to receiving your revised manuscript.

Kind regards,

Jamie Males

Editorial Office

PLOS ONE

Journal Requirements:

MM was supported by a consulting contract with the Johns Hopkins University Applied Physics Laboratory

Reviewers' comments:

Reviewer's Responses to Questions

**Comments to the Author**

1. Is the manuscript technically sound, and do the data support the conclusions?

Reviewer #1: Yes

2. Has the statistical analysis been performed appropriately and rigorously? 

Reviewer #1: Yes

3. Have the authors made all data underlying the findings in their manuscript fully available?

Reviewer #1: No

4. Is the manuscript presented in an intelligible fashion and written in standard English?

Reviewer #1: Yes

5. Review Comments to the Author

Reviewer #1: This paper introduces ideas from stochastic population biology and statistical physics to describe the properties of two broad kinds of cyber systems – (1) the authors consider that each of the N0 components can be in only one of two states (functional or nonfunctional) and they model this situation as a Markov process that describes the transitions between functional and non-functional states; and (2) they consider the situation in which the Operating System (OS) of cyber components is updated in time, they analyze a temporal schedule of OS updates and the probability of transitioning from the current OS to a more recent one via stochastic simulation to capture the pattern of the illustrative data and then derive the forward equation for the OS of a computer at any time. The work is interesting, could prove instrumental for other research contributions in the field and so the reviewer has the following comments and suggestions for improvement:

1) When it comes to variability, in computing systems there are more than two types like the natural and anthropogenic variability. For example, in “Stochastic communication: A new paradigm for fault-tolerant networks-on-chip” VLSI design 2007 (2007) there is a discussion of transient, intermittent and permanent faults. What is not captured in this variability discussion is the transient version when a component can fail due to a soft error or a just one time error caused by some physical phenomenon but then the cyber component behaves as it should. This is classical in satellite communication due to solar cosmic radiation but it can also take place at sea level du to particle hits, packaging nonidealities or lithography manufacturing errors in the CMOS cyber systems.

2) For the discussion of the systems analyzed on pages 4-6 it would be educational and instructive to provide some schematic diagrams to better understand the setup and the developed mathematical analysis. This should also reflect the Markov chain setup, types of states, transition probabilities etc. It is crucial for a reader to understand how the state is defined and can be further extended in other works for other purposes.

3) The prior work of modeling cyber components and cyber-physical systems through concepts from stochastic population biology and statistical physics has been addressed before for various purposes such as modeling communication protocols, fault tolerance, communication / traffic / workloads, buffer sizing, power and thermal management, etc. As the authors can see this has already been considered and actually modeled through a similar formalism in these prior papers

- "Hitting time analysis for fault-tolerant communication at nanoscale in future multiprocessor platforms." IEEE Transactions on Computer-Aided Design of Integrated Circuits and Systems 30, no. 8 (2011): 1197-1210.

- "Statistical physics approaches for network-on-chip traffic characterization." In Proceedings of the 7th IEEE/ACM international conference on Hardware/software codesign and system synthesis, pp. 461-470. 2009.

- "Mathematical modeling and control of multifractal workloads for data-center-on-a-chip optimization." In Proceedings of the 9th International Symposium on Networks-on-Chip, pp. 1-8. 2015.

- "Constructing compact causal mathematical models for complex dynamics." In Proceedings of the 8th International Conference on Cyber-Physical Systems, pp. 97-107. 2017.

and those should be discussed.

4) There is a philosophical issue on whether we should model cyber systems with discrete time or continuous time Markov chains. It would be good if the authors can discuss this as there is also the issue f nonstationarity in these cyber systems possibly also aging, not the biological aging but some form of usage over time. There is also the need for software rejuvenation etc…

5) It would be education to provide the mathematical derivations for all equations like the formula in eq (11) in the appendices.

6) Under what transition probabilities the closed formula be derived?

7) Along the same lines please discuss the limitations and how the Markov chain formalism can be extended for nonstationary and non-Markovian situations which has timidly already been started.

8) Can the authors comment on the scalability of the approach? For example Markov chains are notoriously hard due to state explosion. What are other limitations exist and how should be addressed in the future?

6. PLOS authors have the option to publish the peer review history of their article (what does this mean?). If published, this will include your full peer review and any attached files.

Reviewer #1: No

---

## [Author Response · Author response to Decision Letter 0]

1 Nov 2022

Reviewers' comments

Reviewer's Responses to Questions

Comments to the Author

1. Is the manuscript technically sound, and do the data support the conclusions?

Reviewer #1: Yes

Response: We thank the reviewer for an excellent and thorough report. The reviewer clearly understood and appreciated what we have done, and pointed us towards literature that connects with our work.

In the revision, we have cited all of the papers that the reviewer mentions (details below). Furthermore, we have added additional citations to other relevant literature that we found by following the leads given in the papers mentioned by the reviewer.

3. Has the statistical analysis been performed appropriately and rigorously? Reviewer #1: Yes

Response: No response needed

4. Have the authors made all data underlying the findings in their manuscript fully available?

The PLOS Data policy requires authors to make all data underlying the findings described in their manuscript fully available without restriction, with rare exception (please refer to the Data Availability Statement in the manuscript PDF file). The data should be provided as part of the manuscript or its supporting information, or deposited

to a public repository. For example, in addition to summary statistics, the data points behind means, medians and variance measures should be available. If there are restrictions on publicly sharing data—e.g. participant privacy or use of data from a third party—those must be specified.

Reviewer #1: No

Response: In Appendix S1we now include a description of how the data were collected and two Tables that give the raw data and their summaries. In the first code listing in Appendix S4, we give Rscript to generate Fig 2A. The other codes in Appendix S4 reproduce the rest of the results.

5. Is the manuscript presented in an intelligible fashion and written in standard English?

Reviewer #1: Yes

Response: No response needed, but we have also worked with a professional copyeditor to ensure maximum clarity.

5. Review Comments to the Author

Reviewer #1: This paper introduces ideas from stochastic population biology and statistical physics to describe the properties of two broad kinds of cyber systems – (1) the authors consider that each of the N0 components can be in only one of two states (functional or nonfunctional) and they model this situation as a Markov process that describes the transitions between functional and non-functional states; and (2) they consider the situation in which the Operating System (OS) of cyber components is updated in time, they analyze a temporal schedule of OS updates and the probability of transitioning from the current OS to a more recent one via stochastic simulation to capture the pattern of the illustrative data and then derive the forward equation for the OS of a computer at any time. The work is interesting, could prove instrumental for other research contributions in the field and so the reviewer has the following comments and suggestions for improvement:

Response: Once again, we thank the reviewer for this excellent summary of our paper and its goals.

1) When it comes to variability, in computing systems there are more than two types like the natural and anthropogenic variability. For example, in “Stochastic communication: A new paradigm for fault-tolerant networks-on-chip” VLSI design 2007 (2007) there is a discussion of transient, intermittent and permanent faults. What is not captured in this variability discussion is the transient version when a component can fail due to a soft error or a just one time error caused by some physical phenomenon but then the cyber component behaves as it should. This is classical in satellite communication due to solar cosmic radiation but it can also take place at sea level du to particle hits, packaging nonidealities or lithography manufacturing errors in the CMOS cyber systems.

Response: This is a good point. In addition to citing the paper mentioned by the referee, we spent time reading additional literature and now begin the abstract “Variability is inherent in cyber systems” and the paper with “Although one wishes it were otherwise, variability is a constitutive property of cyber systems” with 5 supporting references. We eliminated the sentence about classification into natural and anthropogenic.

2) For the discussion of the systems analyzed on pages 4-6 it would be educational and instructive to provide some schematic diagrams to better understand the setup and the developed mathematical analysis. This should also reflect the Markov chain setup, types of states, transition probabilities etc. It is crucial for a reader to understand how the state is defined and can be further extended in other works for other purposes.

Response: This is an excellent suggestion. We added a new Figure 1, showing a visual representation of the first problem that we solve (systems with only two possible states). The new Figure 2 includes the data shown in the previous Figure 1, and shows a visual representation of the second problem that we solve (systems with multiple states that are updated unidirectionally).

3) The prior work of modeling cyber components and cyber-physical systems through concepts from stochastic population biology and statistical physics has been addressed before for various purposes such as modeling communication protocols, fault tolerance, communication / traffic / workloads, buffer sizing, power and thermal management, etc. As the authors can see this has already been considered and actually modeled through a similar formalism in these prior papers - "Hitting time analysis for fault-tolerant communication at nanoscale in future multiprocessor platforms." IEEE Transactions on Computer-Aided Design of Integrated Circuits and Systems 30, no. 8 (2011): 1197-1210.

- "Statistical physics approaches for network-on-chip traffic characterization." In Proceedings of the 7th IEEE/ACM international conference on Hardware/software codesign and system synthesis, pp. 461-470. 2009.

- "Mathematical modeling and control of multifractal workloads for data-center-on-a-chip optimization." In Proceedings of the 9th International Symposium on Networks-on-Chip, pp. 1-8. 2015.

- "Constructing compact causal mathematical models for complex dynamics." In Proceedings of the 8th International Conference on Cyber-Physical Systems, pp. 97-107. 2017.

and those should be discussed.

Response: We thank the reviewer for pointing us towards this excellent work, which we now cite in the introduction to the paper (lines 16-20 of the new version) and at various points of contact throughout the paper. We have also added other relevant literature citations as appropriate and thank the reviewer for the inspiration to broaden the reference list.

However, these papers are generally pretty technical – note that they appear in top-rate engineering conferences and use sophisticated tools such as fractional calculus and multidimensional partial differential equations. One goal of our paper is to reach population biologists to encourage them to think about cyber systems, and for this reason we opted for a slightly lower level of technical sophistication. In a real sense, our paper provides an entry point for those mentioned by the reviewer.

4) There is a philosophical issue on whether we should model cyber systems with discrete time or continuous time Markov chains. It would be good if the authors can discuss this as there is also the issue of nonstationarity in these cyber systems possibly also aging, not the biological aging but some form of usage over time. There is also the need for software rejuvenation etc...

Response: We added a section (lines 78-100) in which we discuss characterizing time (and also cite the reviewers work (references 2, 11, 12, and 19) at this point. In addition, Supplement S2 Connecting continuos and discrete time has a more mathematical treatment of the subject; in it we show how to derive continuous time versions of Eqns 3-5. Elsewhere in the paper (lines 208- 211 and 309-313) we discuss that were continuous time used, we would have to deal with differential (in time)-difference (in state) equations that generally require either numerical solution or, in some cases asymptotic solutions (references 33, 38 added).

5) It would be education to provide the mathematical derivations for all equations like the formula in eq (11) in the appendices.

Response: We have rewritten the text (lines 196-205) that lead Eqn 11, since we believe that it is important for readers to be able to follow the logic of this derivation.

6) Under what transition probabilities the closed formula be derived?

Response: We assume that the reviewer is referring to Eqns 14 and 15. In the Discussion we added a section on non-binomial transitions (lines 429-450) and explain that as long as the mean and variance of the transition process can be described by a closed formula, the Gaussian approximation that we develop can be applied.

7) Along the same lines please discuss the limitations and how the Markov chain formalism can be extended for nonstationary and non-Markovian situations which has timidly already been started.

Response: We added a new section (lines 451-462) entitled Non-Markov transition processes. In this section we explain that although non-Markov processes are common in the natural world, scientists and engineers try to find ways to make them Markov. We give two classic cites (references 40, 41). We then give a specific example of how a non-Markov transition process can arise for the first model in which there are just two states and explain how this could be made into a Markov-transition process, giving two additional cites (references 42, 43)

8) Can the authors comment on the scalability of the approach? For example Markov chains are notoriously hard due to state explosion. What are other limitations exist and how should be addressed in the future?

Response: We have made it clear (lines 224-225, 371-366) that the Gaussian approximation we develop for the situation of only two states for the cyber components can be scaled up (as can the simulation). As the number of cyber components in the system increases, the solutions of both forward equations (Eqns 11 and 17) will be limited by computational time, but simulation of the processes will not suffer as greatly.

---

## [Editor Report · Decision Letter 1]

1 Dec 2022

Population Processes in Cyber System Variability

PONE-D-22-10441R1

Dear Dr. Mangel,

We’re pleased to inform you that your manuscript has been judged scientifically suitable for publication and will be formally accepted for publication once it meets all outstanding technical requirements.

Kind regards,

Paul Bogdan

Guest Editor

PLOS ONE

Additional Editor Comments (optional):

After reading the revised manuscript and the responses of the authors I am convinced that this paper is ready for publication. I can see that the authors have gone beyond my comments and enhanced the manuscript significantly so I recommend it s acceptance.

Thank you very much for all their hard work, I think this is a good contribution and as the authors state this paper will attract more attention from biologists to think about cyber systems as good models for biological systems as well.
---

## [Editor Report · Acceptance letter]

14 Dec 2022

PONE-D-22-10441R1 

Population processes in cyber system variability 

Dear Dr. Mangel:

I'm pleased to inform you that your manuscript has been deemed suitable for publication in PLOS ONE. Congratulations! Your manuscript is now with our production department. 

Kind regards, 

on behalf of

Professor Paul Bogdan 

Guest Editor

PLOS ONE